# Childhood socioeconomic position relates to adult decision-making: Evidence from a large cross-cultural investigation

**Simon B. Wang**[1]☯, **Jamie L. Hanson**[1,2]☯*

**1** Learning, Research & Development Center, University of Pittsburgh, Pittsburgh, PA, United States of America, **2** Department of Psychology, University of Pittsburgh, Pittsburgh, PA, United States of America

☯ These authors contributed equally to this work.
* jamie.hanson@pitt.edu

**Data Availability Statement:** The original authors (Ruggeri et al.) posted all data on the Open Science Framework (at https://osf.io/njd62). Additionally, our code is located at: https://osf.io/d578u.

## Abstract

Early exposure to poverty may have profound and enduring impacts on developmental trajectories over the lifespan. This study investigated potential links between childhood socioeconomic position, recent economic change, and temporal discounting in a large international cohort (N = 12,951 adults from 61 countries). Temporal discounting refers to the tendency to prefer smaller immediate rewards over larger rewards delivered after a delay, and connects to consequential outcomes including academic achievement, occupational success, and risk-taking behaviors. Consistent with multiple theories about the impacts of stress exposure, individuals who reported lower socioeconomic positions in childhood exhibited greater temporal discounting in adulthood compared to peers who did not. Furthermore, an interaction emerged between childhood socioeconomic position and recent economic change, such that the steepest temporal discounting was found among those from lower childhood socioeconomic positions who also recently experienced negative economic change as a result of the COVID pandemic. These associations remained significant even when accounting for potentially confounding factors like education level and current employment. Findings provide new evidence that childhood socioeconomic position relates to greater temporal discounting and steeper devaluation of future rewards later in adulthood, particularly in response to contemporaneous economic change. This suggests childhood socioeconomic position may have longer-term impacts on developmental trajectories. Speculatively, childhood socioeconomic position may shape adult behavior through increased life stress, diminished access to resources, and lower perceived trust and reliability in social systems. These findings underscore the long-term implications of socioeconomic gaps, cycles of disadvantage and economic marginalization.

## Introduction

Socioeconomic position and exposure to poverty can significantly shape human well-being and health, as multiple meta-analyses find lower childhood socioeconomic position relates to

**Funding:** This work was supported by internal startup funds provided to Dr. Hanson by the University of Pittsburgh and the Learning, Research, and Development Center. Also, Simon Wang was supported by the University of Pittsburgh David C. Frederick Honors College and DeVito-Lipner Family Fund for Undergraduate Research.

**Competing interests:** The authors have declared that no competing interests exist.

poorer physical and mental health in adulthood [1,2]. This is perhaps not surprising given that lower socioeconomic position often includes food insecurity, unreliability in shelter, family separation and turmoil, and high levels of stress exposure [3–5]. The magnitude of this issue is profound as over 11 million children in the United States, and 356 million children globally, live in poverty [6]. This has led to an expanded focus on developmental processes influenced by socioeconomic position, with a notable interest in decision-making. Decision-making encompasses a complex interplay of cognitive processes and value-based choices [7,8], but one major area of research in decision-making has been temporal discounting; this is the tendency for an individual to devalue future rewards in favor of smaller, more immediate ones. However, there are major knowledge gaps and limited data about the relations between childhood socioeconomic position and this type of decision-making in adulthood.

Independent of socioeconomic position, temporal discounting is connected to important indicators of health [9], financial outcomes [10], impulsivity, and risk behaviors [11,12]. Temporal discounting is also altered in multiple forms of psychopathology [13,14]. For example, individuals who suffer from substance abuse disorders, gambling, and overeating show greater preferences for immediate rewards than those without those issues [15–18]. Understanding the developmental factors that influence individual differences in temporal discounting could help identify people at heightened risk for a variety of negative outcomes, helping to target intervention and prevention efforts. Looking at relations between temporal discounting and socioeconomic position, multiple reports in adults have found associations between high socioeconomic position and a greater ability to delay immediate rewards [19–21]. This has been found in large cohorts (N>40,000) [22] and in multiple registered reports [23]. Similar patterns have also been noted earlier in development, with higher socioeconomic position relating to lower discounting (or greater tendency to delay receipt of immediate rewards) in a sample of over 4,000 children [24].

Important to highlight, past projects focused on temporal discounting and socioeconomic position, continually find higher socioeconomic position is related to greater ability to delay immediate rewards (lower temporal discounting) at multiple points throughout the lifespan. However, few studies, to our knowledge, have (a) looked at the relation between childhood socioeconomic position and adult temporal discounting; (b) examined if temporal discounting is related to the interaction of childhood socioeconomic position and significant economic changes in adulthood; and (c) probed whether these effects are seen in multiple countries and cultures, specifically non-Western, Educated, Industrialized, Rich and Democratic (W.E.I.R. D.) samples. Here, we seek to fill important knowledge gaps related to these issues, by probing associations between childhood socioeconomic position and temporal discounting later in life.

A focus on childhood socioeconomic position and temporal discounting later in life is motivated by two critical bodies of literature that posit significant influences of childhood socioeconomic position on later behavior. First, the developmental origins of health and disease hypothesis emphasizes the persistent influence of the childhood environment. Consistent with this hypothesis, the effects of socioeconomic position are often long-term in nature, as multiple studies have found childhood socioeconomic position is related to adult outcomes, including poorer self-rated health, more chronic health conditions, depressive symptoms, and poorer cognitive functioning in adulthood [25–28]. Notably, physical and mental health disparities fall along a socioeconomic gradient, meaning those at the highest positions enjoy better health outcomes than even those just below them [29,30]. This has been found for both physical and mental health. Related to the developmental origins theory, the graded association of health and disease can begin before birth and compounds over the course of development, reflecting altered biological and psychosocial responses to environmental stressors more commonly experienced moving down the socioeconomic hierarchy [31,32]. Connected to

these ideas, we hypothesized a main effect of childhood socioeconomic position on temporal discounting such that those from lower socioeconomic backgrounds would show greater tendencies to choose immediate over delayed rewards.

Second, stress sensitization models argue that adverse experiences early in development can affect how individuals respond to later challenges, lowering the threshold necessary to trigger a stress response [33]. In other words, stressful childhood experiences, such as those common to individuals living at lower socioeconomic positions, can increase reactivity to subsequent stressful events, increasing the likelihood of negative consequences. Compared to those without, those who experienced adversity during childhood are shown to be at greater risk of depression, bipolar disorder, anxiety disorders, and substance abuse related outcomes, as well as internalizing and externalizing psychopathology, more generally, in response to adult stressors [34,35]. More broadly, stressful experiences in childhood have also been shown to consolidate negative cognitive schemas and dysfunctional attitudes that shape the interpretation of subsequent stressors, increasing vulnerability to stress in adulthood [36,37]. These cognitive biases could relate specifically to financial stress. During the COVID-19 pandemic, the global economy fell into a financial crisis. Unemployment increased around the world, leaving millions without adequate income for food, housing, and other basic necessities [38]. Consistent with stress sensitization theories, we hypothesized that individuals who grow up in lower socioeconomic position households are more likely to respond negatively to recent changes in economic status such as those experienced during the pandemic.

Guided by these ideas, we leveraged a large, cross-cultural secondary dataset [39] to examine the relation between childhood socioeconomic position and adult temporal discounting. This diverse, multinational sample allowed us to examine the longevity and generalizability of potential linkages, as well as interactions between childhood socioeconomic position and recent economic change on adult temporal discounting. Motivated by past work finding long-term links with socioeconomic position and behavior (and developmental origins of health and disease), we first predicted that lower childhood socioeconomic position would be related to greater temporal discounting of future rewards. Second, and guided by the idea of stress sensitization, we also hypothesized that adult temporal discounting would be related to the interaction of childhood socioeconomic position and recent economic change. Finally, given recent research finding country-level factors can shape aspects of decision-making, with greater discounting seen in countries with lower incomes, lower gross domestic products, and greater economic inequality [39], we were interested in exploring if any results vary based on country and non/W.E.I.R.D. samples.

## Methods

### Participants

Data from 12,951 adult participants (47% Female; Mean Age = 34.02) between the ages of 18–89 years were analyzed from a public access dataset collected by other researchers focused on temporal discounting and economic inequality [39]. The original study involved randomly sampling adults from 61 countries with testing completed online via Qualtrics and was given ethical approval by the Institutional Review Board at Columbia University. Participants provided informed written consent remotely at the start of the survey. Full sample demographics are noted in Table 1. For an in-depth description of the study design, recruitment strategy, and general sampling philosophy used in data collection please refer to Ruggeri et al. [39] and https://osf.io/njd62. Of note, our hypotheses were not pre-registered.

**Table 1. Demographic and descriptive statistics of the sample including age, gender, education completed, income, and employment status.**

|  | Overall<br>(N = 12951) |
| --- | --- |
| **Age** | |
| Mean (IQR) | 34.03 (25, 40) |
| Median (Min, Max) | 31.0 (18, 89) |
| **Gender** | |
| Female | 6217 (48%) |
| Male | 6734 (52%) |
| **Education Completed** | |
| Bachelor | 5125 (40%) |
| Primary ed. | 184 (1.4%) |
| Secondary ed. | 2099 (16%) |
| Technical ed. | 1422 (11%) |
| Graduate | 4087 (32%) |
| Missing | 34 |
| **Income (USD Adjusted)** | |
| Mean (IQR) | 27660 (914, 35714) |
| Median (Min, Max) | 7350 (0, 4285710) |
| | |
| **Employment** | |
| Employed | 9383 (72%) |
| Unemployed | 3567 (28%) |
| Missing | 1 |

## Procedure

**Childhood socioeconomic position.**   Childhood socioeconomic position was measured by participants' self-report responses to a survey question asking them to assess their financial circumstances growing up. Specifically, they were asked, "*As a child, how would you describe the financial situation in your household compared to a typical home where you grew up*?" Possible responses included the following: poor, below average but not poor, around average, above average but not wealthy, or wealthy. Responses were then coded on a 5-point Likert scale with "*poor*" corresponding to 1 and "*wealthy*" corresponding to 5.

**Recent economic change.**   Similar to childhood socioeconomic position, recent economic change was measured subjectively by asking participants to reflect on their recent financial status. The question asked, "*Overall, and considering the global pandemic, how did your financial situation change during 2020*?". Responses included, "*My financial situation became much worse*", "*My financial situation became somewhat worse*", "*There were no major impacts on my financial situation*", "*My financial situation became somewhat better*", or "*My financial situation became much better.*" Responses were coded on a 5-point Likert scale with "*My financial situation became much worse*" corresponding to 1 and "*My financial situation became much better.*" corresponding to 5.

**Temporal discounting.**   To assess temporal discounting, participants were presented with a series of hypothetical, adjusting binary choice sets framed across three different baseline scenarios (gain-frame, loss-frame, and high-magnitude gain-frame) followed by four additional questions about specific choice anomalies (present bias, sub-additivity, delay-speedup). A large body of research clearly shows that the framing of choices can affect decision-making [40,41], but most studies on temporal discounting have largely relied upon narrowly defined

measurement techniques that do not richly account for differences in how choices are framed. While the present study is aimed primarily at understanding temporal discounting, it is important to acknowledge the effect of framing and other biases on this type of decision-making. Therefore, we adopted the temporal discounting measurement and operationalization used by Ruggeri et al. [39]; this approach included choices across different frames and represents a more robust overall indication of discounting behavior.

Each scenario consisted of three choice sets between a smaller, immediate option and a larger, delayed option. The initial immediate value was approximately 10% of the national household income average (either median or mean, depending on the local standard) of the participant's country of residence. All participants began by choosing between this value or 110% of it in 12 months (i.e. $500 vs. $550 for US participants). Subsequent choice sets were based on the individual's initial decision. Participants who preferred the immediate gain in the first scenario were presented with the same choice set but with an increased delayed value of 120%. If they again preferred the immediate option, the delayed value again increased to 150%. Participants who preferred the delayed gain in the first scenario were shown a decreased delayed option of 102%. If they again preferred the delayed option, the delayed value decreased again to 101%.

Then, the same progression of choices was inverted and framed as losses rather than gains (i.e. "*If you had to pay for something, which option would you prefer*?"). In this scenario, participants who first chose the immediate option were subsequently presented with increased delayed values of 120% and 150%. Similarly, participants who initially chose the immediate option were shown decreased delayed options of 102% and 101%. Finally, the original gain set progression was repeated using 100% of the national household average income to represent higher-magnitude choices.

Finally, four additional questions were asked to understand specific choice anomalies. All participants were presented with a present bias scenario involving a choice between receiving the base immediate value in 12 months or the maximum delayed value the participant was presented with in 24 months. Next, they were given a subadditivity scenario involving a choice between receiving the base immediate value now or the maximum delayed value the participant was presented with in 24 months. Following this sequence, participants were presented with two scenarios involving delay or speedup of rewards. One scenario was framed as an incentive for waiting (termed a bonus) and the other was framed as a diminishment in receiving the gain earlier.

Based on an individual's choices across all three discounting scenarios and the four additional questions about choice anomalies (present bias, subadditivity, delay, and speedup), an overall temporal discounting score (TD score) was constructed for each participant by assigning a sub-score for each question and adding sub-scores across all questions. Sub-scores ranged from 0–5 for the three discounting scenarios (high-magnitude gain, low-magnitude gain, and loss) and 0–1 for the additional questions that were summed to produce a total TD score ranging from 0 (always prefer delayed gains or earlier losses) to 19 (always prefer immediate gains or delayed losses). Every time the smaller, sooner option was chosen, the participant received 1 point. The TD score measured the extent to which a participant chooses the smaller, sooner option and represents the consistency of temporal discounting, above and beyond other choice anomalies.

Here, we focus on this broader conceptualization of TD behavior because the role of specific choice anomalies falls beyond the scope of our analysis. The purpose of including questions related to these choice anomalies in the overall TD score is to allow for the comparison of discounting irrespective of the presence of other choice anomalies as well as across individuals. Of note, this data was previously collected by Ruggeri et al. [39], and specific choices about

discounting scenarios and subsequent choice sets were made by those investigators. More information about the TD task and reliability, validity, and calculation of the TD score can be found in our supplement.

**Covariates.**   Participants reported their gender as male, female, or other. Participants in the "*other*" category were excluded from our analysis due to little data (less than 3% of the sample). Employment status was coded such that "*unemployed*" included those looking for work, out of work for personal reasons, full-time students, and retired individuals. Those who reported having full-time, part-time, or self-employed work were coded as employed. The "*education completed*" variable categorizes responses to the question "*What is your highest level of education completed*?"

## Statistical modeling

Simple bivariate correlations were run between all quantitative variables to inform model construction. Across our sample, we find modest negative correlations between temporal discounting and measures of current socioeconomic position, namely, income (r = -0.108), recent economic change (r = -0.153), and childhood socioeconomic position (r = -0.022). We also find modest positive correlations between childhood socioeconomic position (r = 0.142) and income (r = 0.148) with recent economic change. Age was also positively associated with income (r = 0.123).

Linear mixed-effects models (LMEMs) were then constructed to examine the main effect of childhood socioeconomic position. We included temporal discounting as the dependent variable, and gender, age, employment status, childhood socioeconomic position, and recent economic change as the independent variables. Gender and age were added to control for any systematic demographic differences in temporal discounting. Employment status was included to account for differences in temporal discounting associated with job security. Both childhood socioeconomic position and recent economic change were added to locate a main effect of childhood socioeconomic position on temporal discounting above and beyond contemporaneous economic shocks and boons.

Careful consideration was given to the use of linear mixed-effect models. Exploratory analyses showed variation in the distribution of temporal discounting (S1 Fig) as well as in both the slope and intercept of the relation between childhood socioeconomic position and temporal discounting across countries (S2 Fig). Thus, we included country of residence as a random effect in our LMEMs. In doing so, our models allow both the slope and intercept to vary by country. Because the sample was highly educated, we conducted additional sensitivity analyses including and withholding education from our model. Next, we planned to look at the interaction of childhood socioeconomic position and recent economic change in predicting temporal discounting. All continuous variables were converted to z-scores. For all significant interaction terms, we conducted simple slope analyses, comparing groups of individuals at the mean recent economic change score, above the mean (+1 SD), or below the mean (-1 SD).

Additionally, our supplementary materials contain sensitivity/complementary analyses to explore the robustness of our effects. These include the following: (i) examination of temporal discounting score distributions between countries, (ii) examination of the simple correlation between childhood socioeconomic position and temporal discounting score between countries, (iii) analysis of reconstructed main effects and interaction models with the addition of global extreme poverty rates in each participant's birth country and year, (iv) analysis of reconstructed main effects models with the addition of a dummy variable indicating whether a country is W.E.I.R.D or not, (v) a multiple regression model examining the relation between temporal discounting and childhood socioeconomic position between countries, and (vi)

reexamination of main effects using individual perceptions of childhood socioeconomic position as a categorical indicator, rather than a continuous variable.

## Results

### Probing associations between childhood socioeconomic position and temporal discounting

Our first linear mixed effects model tested for a main effect of childhood socioeconomic status on temporal discounting, including participants' current employment status, age, and gender as covariates. As shown in the first column of Table 2, temporal discounting was negatively related to childhood socioeconomic status ($\beta$ = -0.02, t = -2.78, p = 0.005, d = -0.05), meaning those who reported lower childhood socioeconomic positions exhibited greater preference for immediate rewards (i.e. higher temporal discounting scores). Sensitivity analyses including education level as a covariate yielded similar results, though with associations slightly attenuated (See the second column of Table 2 and in Fig 1). Temporal discounting was related to individual perceptions of childhood socioeconomic status ($\beta$ = -0.02, t = -2.14, p = 0.03, d = -0.04).

In line with the modest negative correlations found between temporal discounting and childhood socioeconomic status, we found the fixed effects of these two models to explain similar proportions of the variance in the data. The first model explains about 0.5% of the variance in temporal discounting, and the second explains about 0.8%. However, in looking at the variance explained by both fixed and random effects, we found the second model accounted for about 22.3% of the variance in temporal discounting, whereas the first model only accounted for about 2.16% of the variance in temporal discounting. This difference is likely due to the correlation between education and country of residence.

The model on the left considers the main effect of childhood socioeconomic position on delay discounting. The model on the right side considers the same main effect while controlling for education completed. Other than education, both models included the same covariates

**Table 2. Main effects of childhood socioeconomic position on temporal discounting with and without education-level as a covariate.**

| Predictors | Temporal discounting | | | Temporal discounting | | |
|---|---|---|---|---|---|---|
| | *Estimates* | *CI* | *p* | *Estimates* | *CI* | *p* |
| (Intercept) | 0.00 | -0.13–0.10 | 0.959 | 0.00 | -0.12–0.13 | 0.950 |
| Employment (Unemployed) | 0.03 | 0.05–0.15 | 0.059 | 0.01 | -0.03–0.04 | 0.765 |
| Age | 0.03 | 0.00–0.04 | **<0.001** | 0.04 | 0.03–0.06 | **<0.001** |
| Gender (Male) | 0.02 | -0.01–0.05 | 0.300 | 0.01 | -0.02–0.04 | 0.586 |
| Childhood Socioeconomic Position | -0.02 | -0.04- -0.01 | **0.005** | -0.02 | -0.03 –-0.00 | **0.032** |
| Education Completed (Primary ed.) | | | | 0.08 | -0.05–0.21 | 0.248 |
| Education Completed (Secondary ed.) | | | | 0.12 | 0.07–0.17 | **<0.001** |
| Education Completed (Technical ed.) | | | | 0.04 | -0.01–0.10 | 0.107 |
| Education Completed (Graduate) | | | | -0.05 | -0.08 –-0.01 | **0.022** |
| **Random Effects** | | | | | | |
| $s^2$ | 0.76 | | | 0.76 | | |
| $t_{00}$ | 0.22 Country | | | 0.22 Country | | |
| ICC | 0.22 | | | 0.23 | | |
| N | 61 Country | | | 61 Country | | |
| Observations | 12950 | | | 12916 | | |
| Marginal $R^2$ / Conditional $R^2$ | 0.002 / 0.225 | | | 0.005 / 0.231 | | |

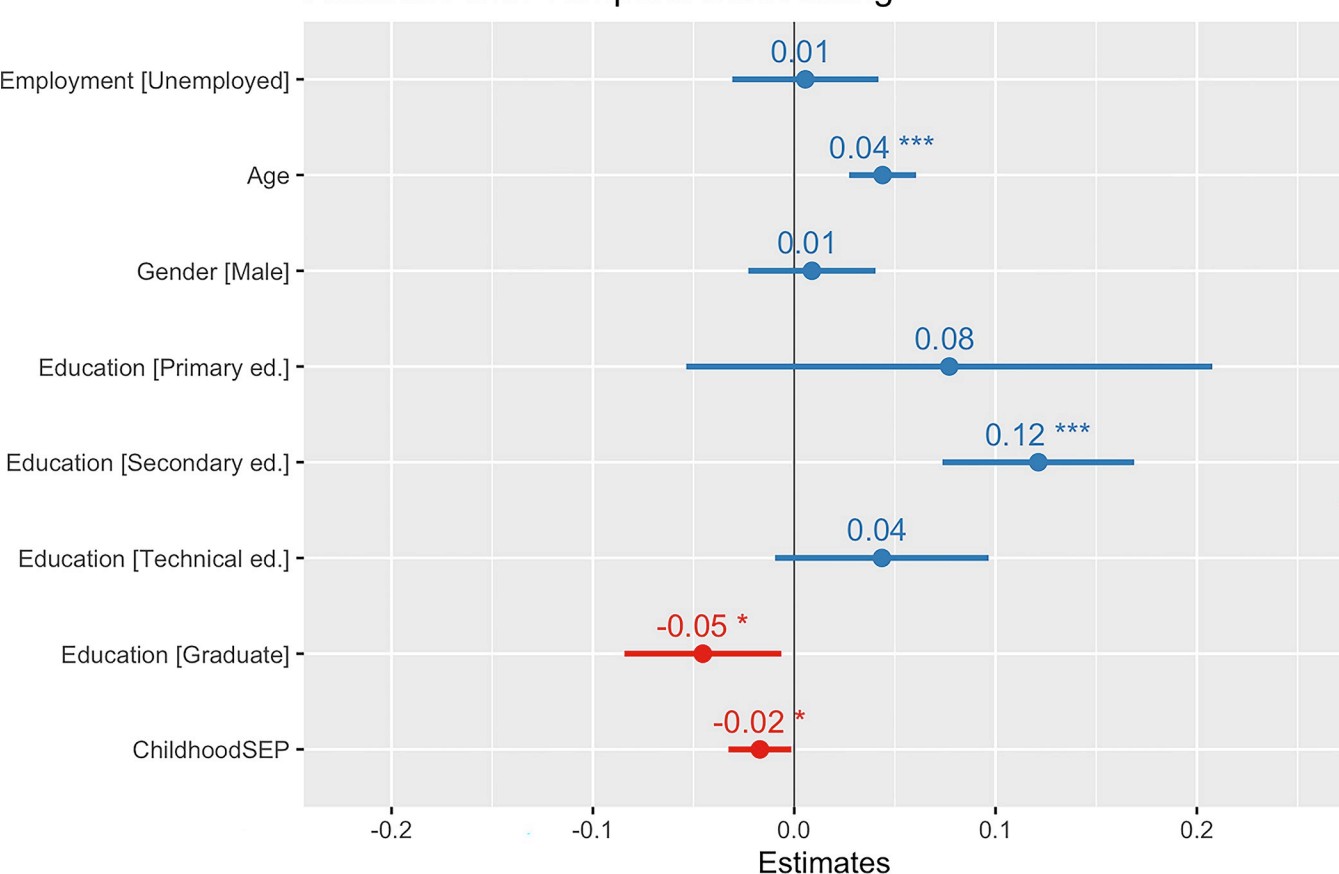

**Fig 1. The coefficient output of our first linear mixed effects model considers the effect of individual-level childhood socioeconomic position on discounting scores controlling for employment status, age, and gender, including country as a random effects term.**

(current employment status, age, and gender) and a random effect of participants' country of residence. Bolded values indicate effects that were significant at p<0.05. Notably, the main effect of childhood socioeconomic position remained significant even when education was added to the model as a covariate.

## Probing associations between interactions and temporal discounting

Next, we examined the interaction effect of childhood socioeconomic position and recent economic change on temporal discounting. The results of the model show significant main effects of childhood socioeconomic position (β = -0.015, t = -1.87, p = 0.06, d = -0.05) and recent economic change (β = -0.07, t = -8.65, p<0.0001, d = -0.15) as well as a significant interaction effect between the two (β = 0.017, t = 2.23, p = 0.02, d = 0.04). This is shown in Fig 2. Follow-up simple slope analyses were conducted using the *interaction* library in R to further examine significant interaction terms.

Results indicate, that for individuals whose childhood socioeconomic position was either average or below average, decreases in recent economic change were significantly associated with higher temporal discounting scores (average childhood socioeconomic position β = -0.07, t = -8.66, p<0.01 below-average childhood socioeconomic position β = -0.09, t = -8.27, p<0.01). In other words, those who experienced more negative economic changes as a result

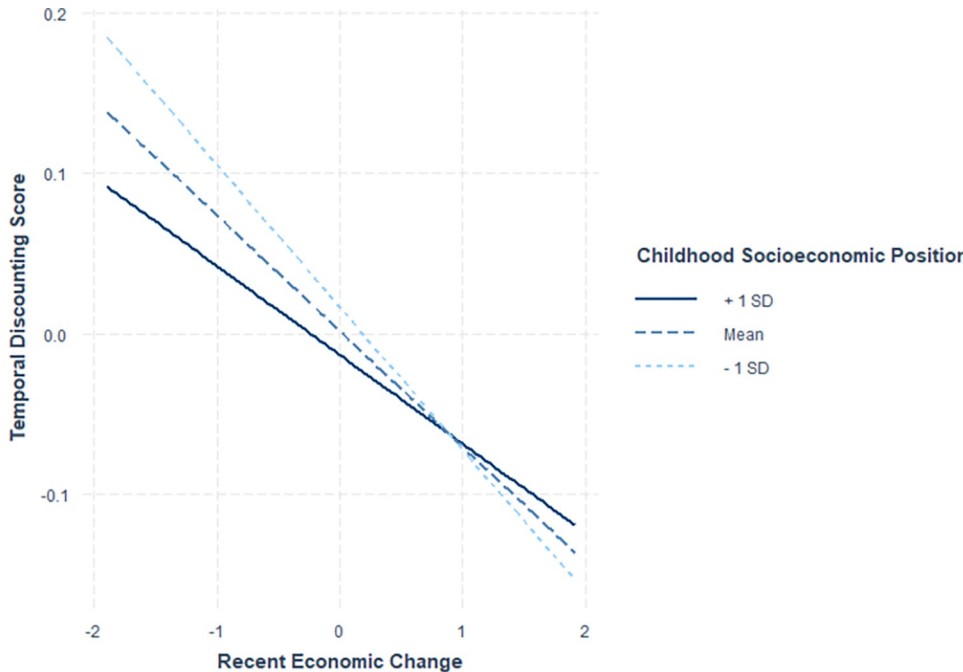

**Fig 2. The association between recent economic change and discounting scores in adulthood at three levels of childhood socioeconomic position.** Those reporting higher (+1 SD), mean, and lower levels (-1 SD) of recent economic change are all shown above. Recent economic change is on the horizontal axis and adulthood temporal discounting scores are on the vertical axis.

of the pandemic showed greater preferences for immediate rewards. There was no effect of childhood socioeconomic position on TD for those with above-average recent economic change (above-average childhood socioeconomic position $\beta$ = -0.06, t = -4.86, p<0.01).

## Discussion

Here, using a large, multinational sample, we assessed relations between childhood socioeconomic position and one form of decision-making, temporal discounting, in adulthood. We were interested in how this early life factor is directly and interactively related to temporal discounting of hypothetical rewards. As we predicted, lower childhood socioeconomic position was related to a greater preference for more immediate rewards. However, and of note, the results reported, though statistically significant, have small effect sizes that indicate childhood socioeconomic position explains only a modest proportion of variance in adult temporal discounting. We also found a significant interaction between childhood socioeconomic position and recent economic change, suggesting that as perceptions of childhood socioeconomic position increase, the effect of recent economic change on temporal discounting decreases. Consistent with stress sensitization theory, those who perceived themselves as lower socioeconomic position in childhood and also reported a recent negative economic change were the most likely to prefer more immediate, smaller rewards compared to larger, more delayed ones, indicating a poorer response to recent stress. Sensitivity analyses, controlling for multiple potential confounds and using childhood socioeconomic position as a discrete rather than continuous variable, suggested similar effects of childhood perceptions of socioeconomic position on temporal discounting. Examined collectively, the longevity of these links is important, and

underscores how socioeconomic position and poverty shape human well-being and health by influencing responses to economic stress.

Our results build upon and connect to past research focused on socioeconomic status, poverty, and decision-making. Previous research has shown a relatively consistent relation between socioeconomic status and decision-making, with evidence linking lower socioeconomic position to greater rates of temporal discounting throughout the lifespan, including in children, adolescents, and adults [42–44]. Of note, many of these past reports were conducted in the United States and other Western, Educated, Industrialized, Rich, Democratic (WEIRD) cultures. In contrast, this project pulls from 61 countries, offering greater generalizability of the reported associations. Critical to highlight, Ruggeri and colleagues [39] found economic inequality and broader financial circumstances were correlated with population choice patterns in this same sample of participants. This, however, was for recent economic change and inequality, compared to the present study which focused on childhood socioeconomic position. Other studies in international samples of adults have examined perspectives about the future, a related measure to temporal discounting, finding greater patience and time preference to be positively associated with high income [45,46]. As such, our findings connect to past projects and suggest both childhood socioeconomic position and economic change in adulthood can affect decision-making.

Findings from this project also connect to the large body of work noting the long-term effects of early socioeconomic position. As noted previously, multiple studies have found childhood socioeconomic position is related to many adult outcomes including poorer self-rated health, more chronic health conditions, depressive symptoms, and poorer cognitive functioning in adulthood. While these relations have been commonly noted for physical and mental health, fewer reports have examined childhood socioeconomic position and adult decision-making. Notably, Ruggeri and colleagues [47] examined a small subsample of this same cohort who grew up in disadvantaged households but then reported above-average financial well-being as adults (termed "*positive deviants*" by the authors). These investigators did not find decision-making was significantly related to cognitive biases for individuals who "*overcame*" low-income childhoods compared to individuals who remained low-income as adults.

In contrast, and in keeping with past work on socioeconomic gradients [29], we analyzed the full range of childhood socioeconomic positions by looking at interaction effects. We did this using both continuous and discrete indicators of childhood socioeconomic position and found relations with temporal discounting. Decision-making processes may continue to be negatively affected by childhood experiences well into adulthood, affecting responses to negative changes in one's economic situation. This may be true even among those whose financial situation has improved since their early years. Furthermore, the interaction effect between childhood and recent economic change paints a more complete picture than analyzing positive deviancy, suggesting that perceptions of socioeconomic position in childhood have enduring effects on decision-making, even if the socioeconomic environment improves over time. Such patterns add to a major body of research linking childhood socioeconomic position to many adult outcomes, expanding from physical and mental health into decision-making.

Thinking about what might be driving the reported relations, there are many possible pathways linking childhood socioeconomic position and adult temporal discounting. First of all, growing up at a lower socioeconomic position may be linked to greater levels of early life stress and environmental challenges. This fits with a large body of work finding that lower socioeconomic position is related to family turmoil, violence exposure, and housing problems or instability. All of these factors may impact development, and future research could explore longitudinal links between childhood socioeconomic position, stress and environmental challenges, and decision-making.

Relatedly, we believe environmental volatility may be critical to investigate. To our knowledge, few projects have examined volatility, unpredictability, and decision-making However, early unpredictability has been linked to poorer mental health and cognition [48]. Interestingly, Kidd, Palmeri, and Aslin [49] found that children who witnessed unreliability in the environment, a form of volatility and unpredictability, were less likely to delay for a later reward in the classic "Marshmallow" test. Finally, perceived childhood socioeconomic position may play an enduring role in decision-making by changing motivation and trust in the environment, including individuals and larger societal systems. Opting for a delayed reward inherently involves trust. Experiences of uncertainty, unreliability, or discrimination such as those associated with poverty are likely to compound over time, diminishing that trust, and leading to preferences for more certain, immediate outcomes. For example, economically marginalized communities often have higher rates of unbanked individuals, even though access to banking is essential to advancing in today's economy [50]. Therefore, those whose sense of trust withstand these negative experiences discount less and may enjoy better outcomes across the board as a result [51–53].

While we believe this work yielded some important results, it is critical to acknowledge several limitations of the present study. For one, using a secondary dataset collected by other researchers limited our access to participant-level observations. Childhood socioeconomic position was reported retrospectively, potentially biasing our results due to memory effects. As mentioned previously, future studies leveraging longitudinal methods to track the incidence of different forms of childhood adversity alongside fluctuations in household socioeconomic position may yield important insights to validate our findings. We were also limited to subjective measures of childhood socioeconomic position and recent economic change. Ideally, we would have also collected other objective measures such as zip code, household income, and parental occupation and education-level. An interesting avenue for future work would be to compare subjective and objective measures of childhood socioeconomic position and recent economic change, probing the differential effects of relative versus absolute socioeconomic experiences [54].

Related to the sample, it is worth noting that while this sample is cross-cultural, it is also highly educated (39.6% have a Bachelor's degree and 31.4% have a graduate degree); this has been a limitation of multiple past studies attempting to compile multinational datasets and not unique to the present study [55,56]. This is an expected result of the survey distribution method (i.e., Qualtrics) which was intentionally varied to avoid bias across countries as well as the online nature of Qualtrics which skews toward populations with internet access. As more work is done to test how representative highly educated individuals are of the world's cultural diversity, we recommend future analyses similarly aimed at generalizable effects to leverage samples with greater diversity in education-levels.

Additionally, using the overall discounting score rather than any single choice set provided a broad view of individual decision-making tendencies, but we also recognize specific choice anomalies may have distinct relations with childhood socioeconomic position. Furthermore, the methodology behind the overall discounting score is relatively rudimentary compared to other advanced approaches for looking at discounting rates. In the future, researchers should consider investigating the unique relations that socioeconomic position has with different forms of decision-making. For example, utilizing a more sophisticated method like Bayesian estimation to model discounting scores could be useful in examining these potential effects. Doing so may elucidate important differences in how the early economic environment affects decision-making and has been shown to be particularly informative.

Unfortunately, poverty remains an all too common experience for children around the world. Historically, policymakers and public officials have often referred to economic mobility

as a sign of strength, representing an opportunity for individuals to escape poverty. Indeed, high economic mobility is often regarded as an indicator of a well-functioning, equitable economy. However, our findings here complicate this traditional understanding, suggesting that challenging socioeconomic environments may continue to influence decision-making well into adulthood, shaping responses to recent economic change. In other words, although economic mobility may translate to more opportunities to improve one's status, the environmental and psychosocial stressors of growing up in poverty have profound negative effects on behavior, health, and overall well-being that may not be completely remedied by achieving higher socioeconomic positions.

We are hesitant to make any specific policy recommendations based on the present study, but generally, our results encourage policies that strengthen the social safety net and narrow gaps between socioeconomic positions. This could limit the concerning pattern of outcomes associated with poverty, especially that of extreme poverty (currently measured as living on less than $1.25 a day). Furthermore, our work highlights the sustained vulnerability of people from low socioeconomic backgrounds to negative economic change, emphasizing the need for support systems during economic downturns. More broadly, our findings contribute to a robust literature outlining the difficulties of growing up in poverty and its lasting effects on human behavior and health.

## Supporting information

**S1 Text. Supplementary materials.**
(DOCX)

**S1 Fig. Distributions of temporal discounting broken down by country.**
(TIF)

**S2 Fig. Relations between childhood socioeconomic position and temporal discounting by country.**
(TIF)

**S3 Fig. Forest plot showing the coefficient estimate for childhood socioeconomic position in every country represented by the sample.**
(TIF)

**S4 Fig. Combined plots showing the discounting score distributions of the five childhood socioeconomic position categories.**
(TIF)

## Acknowledgments

We have no acknowledgments to make at this time.

## Author Contributions

**Conceptualization:** Simon B. Wang, Jamie L. Hanson.

**Data curation:** Simon B. Wang, Jamie L. Hanson.

**Formal analysis:** Simon B. Wang, Jamie L. Hanson.

**Investigation:** Jamie L. Hanson.

**Methodology:** Simon B. Wang, Jamie L. Hanson.

**Resources:** Jamie L. Hanson.

**Software:** Jamie L. Hanson.

**Validation:** Jamie L. Hanson.

**Visualization:** Simon B. Wang, Jamie L. Hanson.

**Writing – original draft:** Simon B. Wang, Jamie L. Hanson.

**Writing – review & editing:** Simon B. Wang, Jamie L. Hanson.

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
