## [Decision Letter · Decision Letter 0]

8 Jan 2024

PONE-D-23-32303Childhood socioeconomic position related to adult decision-making: Evidence from a large cross-cultural investigationPLOS ONE

Dear Dr. Hanson,

Thank you for submitting your manuscript to PLOS ONE. After careful consideration, we feel that it has merit but does not fully meet PLOS ONE’s publication criteria as it currently stands. Therefore, we invite you to submit a revised version of the manuscript that addresses the points raised during the review process.

After thorough evaluation and deliberation, the reviewers have provided their assessments of your manuscript. It is noted that while there is a divergence in their overall evaluation of the paper's quality, there is a consensus regarding the significance of the findings in the context of their 'importance'. The large dataset presented allows for the moderately significant results in various measures. However, there remains an essential query: what does this discovery signify in the broader context of temporal discounting and its correlation with childhood poverty? The reviewers find the results intriguing but emphasize the necessity for a more detailed exposition on why these findings are pivotal and how they address existing gaps or problems in the field.

Furthermore, the second reviewer has articulated several methodological concerns. While these do not inherently impede publication, they certainly merit your careful consideration and response. Addressing these issues could substantially enhance the robustness and credibility of your work.

In light of the above, and considering the valuable insights provided by the reviewers, it is the decision of the handling editor that your manuscript requires major revisions. The implementation of the reviewers' suggestions is anticipated to significantly elevate the quality and impact of your paper. We look forward to receiving your revised manuscript, which we believe has the potential to make a meaningful contribution to the field.

We look forward to receiving your revised manuscript.

Kind regards,

Rei Akaishi

Academic Editor

PLOS ONE

“This work was supported by internal startup funds provided to Dr. Hanson by the University of Pittsburgh and the Learning, Research, and Development Center. Also, Simon Wang was supported by the University of Pittsburgh David C. Frederick Honors College and DeVito-Lipner Family Fund for Undergraduate Research.”

Additional Editor Comments:

After thorough evaluation and deliberation, the reviewers have provided their assessments of your manuscript. It is noted that while there is a divergence in their overall evaluation of the paper's quality, there is a consensus regarding the significance of the findings in the context of their 'importance'. The large dataset presented allows for the moderately significant results in various measures. However, there remains an essential query: what does this discovery signify in the broader context of temporal discounting and its correlation with childhood poverty? The reviewers find the results intriguing but emphasize the necessity for a more detailed exposition on why these findings are pivotal and how they address existing gaps or problems in the field.

Furthermore, the second reviewer has articulated several methodological concerns. While these do not inherently impede publication, they certainly merit your careful consideration and response. Addressing these issues could substantially enhance the robustness and credibility of your work.

In light of the above, and considering the valuable insights provided by the reviewers, it is the decision of the handling editor that your manuscript requires major revisions. The implementation of the reviewers' suggestions is anticipated to significantly elevate the quality and impact of your paper. We look forward to receiving your revised manuscript, which we believe has the potential to make a meaningful contribution to the field.

Reviewers' comments:

Reviewer's Responses to Questions

**Comments to the Author**

1. Is the manuscript technically sound, and do the data support the conclusions?

Reviewer #1: Partly

Reviewer #2: No

2. Has the statistical analysis been performed appropriately and rigorously? 

Reviewer #1: Yes

Reviewer #2: I Don't Know

3. Have the authors made all data underlying the findings in their manuscript fully available?

Reviewer #1: Yes

Reviewer #2: Yes

4. Is the manuscript presented in an intelligible fashion and written in standard English?

Reviewer #1: Yes

Reviewer #2: No

5. Review Comments to the Author

Reviewer #1: The manuscript by Wang and Hanson provide an interesting test of how childhood poverty can influence adult decision making, particularly temporal discounting. The introduction provides a strong rationale for the study, which includes a very large mult-national database to test their ideas. The analyses were quite sophisticated, and in parts a bit challenging to follow for this reviewer without expertise in these methods. The delay discounting task was not the common Kirby questionnaire, or the adjusting amount methodology, but a variant adapted for this research that appeared to attempt to match the monetary values with the differences by income level of the country.

The authors were very clear in the limitations to the study, primarily the self-report nature of the determination of poverty and current financial situation, which can be impacted by self-report bias as well as memory of childhood situation. This may be an issue because previous studies suggests that parents often try to protect their children from experiencing poverty, while they themselves may go without resources.

My main concern about the paper can also be considered a strength. The paper takes advantage of a very large dataset of over 13,000 participants. With a dataset of this size, very small relationships can by significant. Large datasets are great, and the use of a multinational dataset increases the generalizability of the data. However, what was the effect size? How important is this relationship given the very small beta coefficients? This does not detract from the author’s ideas, but I believe it is critical to understand the importance of these previousl experiences to current decision making.

In summary, I really liked this paper, but the authors need to indicate the importance of these results. In the beginning of the results, the authors describe the relationships as modest. That is vague. The important point, if possible, is to describe the amount of variance in current discounting accounted for by the relevant variables. Even if small, this may be very important and definitely worth reporting. I look forward to reviewing a revised version of this stimulating paper.

Reviewer #2: The paper is an interesting discussion of an important topic, but as it stands I am not sure I would recommend the current paper for publication for three reasons:

First, it does not lay out a clear enough theoretical framework through which to guide the research’s approach, predictions, and findings. Although theoretical contribution is not a key criterion for publication in PLOSOne, it does matter to the extent that the questions and the way of approaching them can be scientifically justified and easily interpreted.

• The framing does not clearly enough outline why individual-level decision-making should necessarily be implicated as an important topic of study in the association between experiences of adversity and temporal discounting (as opposed to say structural disadvantage, e.g., lack of access to healthcare), and it does not yet clearly enough argue for why and how experiences of adversity in childhood should affect decision-making, above and beyond current experience.

• Despite framing the gap being addressed as a lack of understanding of the link between SEP and specific decision-making processes, the temporal discounting items used are undertheorized and underexplained – what is different in the gain, loss, and magnitude frames, and why are all three included? Why should we care about current income more so than other economic aspects of financial adversity (as Ruggeri et al., 2022’s paper on temporal discounting indicates we should)? What are the specific ‘choice anomalies’ tested for in the supplemental questions, and what can they tell us above and beyond the primary measures? How were the income titration percentages calculated, and why do they seem to use both participant income and national household average income? Why can we be confident in grouping together all of the measures into a single average (and is an average appropriate – i.e., would a factor analysis allow for a more appropriate weighting of the different components?) More theoretical contextualization is needed to explain why each temporal discounting measure should be included, what we should expect from each, and why they should be combined.

• More attention could be paid in the initial framing of subjective vs. objective measures of childhood SEP. The measure of adult SEP is particularly problematic, as judgment of how things have changed cannot be used as a proxy for judgment of how one is doing overall (as it can make a poor person who did well in the pandemic look like they are higher in SES than a rich person who did poorly in the pandemic). The wording of this measure thus needs to be changed throughout, to something like ‘recent economic change’; alternatively, so that they can speak to something like adult SEP, the authors could use adult income in their analyses instead.

• It seems that one of the key arguments is to push back on Ruggeri’s ‘positive deviance’ findings, but this is not mentioned in the setup. I would love a clearer articulation of the study’s theoretical framework, subsequent predictions, and if/how those predictions might prove important given the context of alternate findings in the literature.

Second, the methods employed are not transparently documented or adequately explained.

• Although preregistration of analysis of secondary data is only now becoming a norm, its absence here makes it harder to both 1) interpret how the analyses link to the theoretical framing, and 2) discount the role of researcher degrees of freedom. (E.g., a global temporal discounting score that combines the sub-variables would be more convincing if this analysis approach had been pre-registered. Without the pre-registration and especially without documentation of the related code readers are left wondering if the results hold only when the sub-items are combined).

• Relatedly, the lack of supplemental information means that the analyses reported cannot be checked, and greater detail on the measures and study sample used cannot be adequately considered.

• Similarly, a greater discussion could be included as to why each method was selected, the reasoning behind how each method was conducted (e.g., more detail on the reasoning behind country-level random effects), and more reporting of missing values (why are missing values imputed?) and related robustness checks.

Third, its contribution to the scientific literature is not clearly outlined.

• As I understand it, the dataset being used was collected by other researchers, however this should be more clearly outlined (and the benefits and limitations discussed). It is also not clear from the original authors’ OSF file whether analysis of this data for use in future publications (as opposed to just for replication of the analyses in their paper) is permitted.

• The use of existing datasets can be sufficient to warrant publication, however the current analyses underleverages the potential of the existing dataset. (E.g., getting beyond WEIRD samples is noted as a benefit in the Discussion section, but no analysis is made to consider if and how results vary according to country context such as country GDP and inequality).

• If the paper is specifically pointing to limitations in existing research (Ruggeri’s ‘positive deviance’ findings), then it could more tightly point to limitations in that existing research and structure the analyses and discussion appropriately – as it stands I worry it is both too specific and too broad to meaningfully contribute above and beyond the original analysis.

• While Figure 3 is compelling, greater discussion is needed to explain differences in the relationship between childhood SEP when measured continuously or discretely (Figure 4).

• As outlined in the discussion, the paper does not dig deeply into underpinning mechanism. As such it is hard to rule out alternative explanations, particularly given the cross-sectional nature of how the data are reported.

Detailed Notes by Section

Abstract

- ‘An interaction emerged between childhood and current socioeconomic status, such that the steepest temporal discounting was found for those facing persistent economic adversity across both time periods.’ – Are we talking about relative adversity or absolute adversity, if we’re comparing across countries?

- ‘Findings provide new evidence that childhood poverty relates to more impulsive decision- making and steeper devaluation of future rewards later in adulthood, despite changes in economic mobility. This highlights poverty's persistent effects on core psychological processes underlying decision-making, suggesting early life socioeconomic position may have longer-term impacts on developmental trajectories.’ Careful with the language here – is it temporal discounting or impulsivity? Do we actually have purchase on psychological mechanism? How does it impact trajectories, and are we sure this is via decision-making?

- ‘By elucidating developmental mechanisms linking childhood poverty to adult well-being, findings could inform policies aimed at narrowing socioeconomic gaps and disrupting cycles of disadvantage and economic marginalization.’ Interesting – I will be looking for more elaboration.

Introduction

- Line 51-52 – I want one more connecting phrase to explain why we should consider the role of decision-making in the connection between socioeconomic position and health

- Fairly broad in potential impacts – there is a tension here between wanting to pin down which types of decision-making are impacted, and drilling into a specific kind of decision-making (temporal discounting). I might pick one.

- Paragraph 2 (line 62) is interesting – I agree that childhood poverty is an important issue, but in some ways this undermines the argument about decision-making. Shouldn’t this issue be important enough that policy-makers take it seriously even if we don’t have clear links to decision-making? If it has such pervasive effects, why should we be interested in decision-making as opposed to other, more structural mechanisms through which poverty might affect outcomes (e.g., access).

- Interesting that in paragraph one there are claims about us not understanding the link between SE position and specific decision-making processes, but then in paragraph 3 (line 73) widens the lens to temporal discounting as a catch-all (despite Ruggeri et al.’s 2022 article which breaks temporal discounting down into more specific components).

- More work could be done in lining up why inequality should be thrown into the mix of potential mechanisms (89) – this comes on line 99, but it is strange to not have it before.

o Here we may need more reasoning as to why we should care about inequality at the level at which it is measured (vs. at a more local level), or more reasoning as to why national-level inequality is the closest proxy we can get in place of more local measures.

o It is unclear what is meant by ‘individual inequality’.

- Line 108 – greater explanation could be given as to what is implicated by stress sensitization, perhaps linking it in clearer language to the study at hand and relevant predictions. is? I would think that the difference would be between those who have experienced stress previously (and become sensitized to it) vs. those that hadn’t? (rather than those would have continued to be in a low SES position vs. Those that have experienced upward mobility)

- Final paragraph of introduction doesn’t go into detail on alternative mechanisms (e.g., the Jachimowicz et al. (2017)’s distinction between can you afford to wait, and do you trust that the thing you are waiting for will come to you?) – are we going to look at mechanism, or just verify the persistence of relationships?

Method

- Participants – remind us of the representativeness of those individuals, particularly in non-WEIRD contexts? Are there limitations there in terms of who has access to Qualtrics (i.e., is this truly a random sample, or is this a broad sample that over-represents higher SES or higher educated individuals from non-WEIRD countries) – the education and student status stats make me wonder if the sample is truly representative, and the income stats (if it is cross-country) should be reported with a clearer unit (e.g., is income adjusted to USD?)

Procedure

- Socioeconomic Position

o Clarify that this does not include an objective measure of SES growing up and currently, but rather subjective measures, and that the current subjective measure is potentially distinct from more general measures of subjective economic position in that it specifically anchored to changes during the pandemic.

o What percentage of the inequality data were imputed? (160) Are there robustness checks excluding the imputed data?

o For the inequality measure, I would like more explanation as to theorized mechanism in guiding which inequality measure was selected (childhood mortality and life expectancy I would expect to act quite differently from GDP or Gini wealth) – which measure here matters, and how is it theorized to impact decision-making?

o Similarly, I lost track of the reasoning behind the imputation and dataset pooling method – why not pick a single measure? What are ‘historical’ measures of ‘global inequality’ – are these measured in the same time period and at the same national level? If a single measure wasn’t picked, more detail is needed as to the reasoning behind the pooling methodology, and how it was conducted (or at least link us to the code).

- Temporal Discounting

o Why the gain, loss, and magnitude frames? This needs more theoretical contextualization to explain why we should include each, and what we should expect from each.

o Why calibrate to the participant’s annual income? Here it would be helpful to either explain the reasoning or to cite someone else who has previously used this methodology.

o What is a TD score? (175) – Total discounting? It might be worth spelling it out the first time it is referenced.

o Similarly, what are the supplemental questions? (175) What do they tell us that we do not already have in the previous scores, what do we predict they will show, and how should they be interpreted? What specifically is meant by a ‘choice anomalies’, and why does analysis of specific anomalies fall beyond the scope of this work? (180) Was this pre-registered?

o How do we access the supplement? (184) Why then on 412 are there ‘no supplementary materials to provide’?

o How was a gain of 10% decided upon? Why a time period of a year? (189)

o Similarly, why 120 then 150? (vs. 102 and 101?) (193)

o Did I understand this correctly, that the discounting choice value was based on participant reported annual income (as specified in line 171), and then the increase and decrease in subsequent questions was based on the national household average income? (187) If so, why? (E.g., are personal and household incomes comparable? Are choices still equivalent at different points along the income scale, where those with lower relative starting incomes will be asked about a greater relative gain vs. those with higher relative starting incomes – i.e., those of lower socioeconomic position are receiving a different question insofar as they are being ask to choose between options with relatively greater gains and losses, as compared to their higher income peers?)

- Statistical Modeling

o Overall – I would recommend bulking up the reasoning behind each analysis. (What question is it meant to answer, and why is the type of analysis selected best? Are there alternative explanations or robustness checks that can be tested?)

o Why use an LMEM model? (Do we expect effects to be linear?)

o Make more explicit why country of residence has a random effect but not the other variables (e.g., age).

o Are these planned analyses (215) pre-registered?

o Why are there multiple analyses, and how do they relate to one another? (What will each of the different analyses give us insight into and are they testing variations on the same hypothesis – i.e., how should we think about correction for multiple hypothesis testing?)

Results

- What do we make of the correlations being so low?

- Table 2: It would be helpful to have a breakdown of the study demographics according to these categories. (e.g., what percentage of the sample listed Gender as Other? What percentage was retired? Etc.)

- Why do a hierarchical regression but only for education completed (and not the other demographic variables like gender and employment?) If we’re interested in Childhood SEP I might run the model looking just at that and then see if it holds when controlling for potential alternative explanations. (But tell us why each of the control variables is important to include).

- What should we make of the extremely small estimates for childhood SEP? How can this be interpreted in real terms? (Even if the association is significant, is it meaningful?)

- For the interaction probe, how is current SEP calculated? (266)

- Figure 3 – if we’re interested in the effects of childhood SEP, I wonder if it wouldn’t be more intuitive to look at current SEP with lines breaking down childhood SEP SDs.

- Explain why is it important to look at childhood SEP categorically. (282) (E.g., Because it is measured on a Likert?)

- Why is the absence of any main effect of country-level inequality justification for not looking for an interactive role for inequality?

Discussion

- The sample is multinational – I was curious to see more breakdown by country GDP, country inequality (especially given the intro about inequality?) etc.

o This breakdown would be particularly useful to inform discussion of WEIRD differences in the Discussion section (311).

- Description of the results of the simple slopes analysis is inadequate: ideally this should break down the interaction to compare either the effect of adult SEP at different levels of childhood SEP or vice versa, not just say which group did the most discounting.

- Careful with the wording (303) – this does not show us childhood socioeconomic position, just current perceptions of childhood position.

- Where does reported financial hardship come in if we have reported annual income? How should we understand differences between subjective report (which might be more susceptible to something like method variance in pessimism) vs. more ‘objective’ measures like reported income? For ‘objective’ measures, what are their strengths and limitations? (e.g., is income the best variable to measure economic situation, particularly given Ruggeri et al.’s (2022) findings about different aspects of financial hardship being important predictors of temporal discounting?)

- Could this work be similarly linked to population choice patterns, but considering childhood rather than current financial status? (313)

- 331 In the discrete version a relationship was not found with discounting- how can this be explained?

- 338 What predictions? These should be more clearly outlined in the setup.

- Careful with what work is cited – The example on 349 doesn’t illustrate the point in 347 (i.e., perceived unsafety vs. perceived safety does not give us clear purchase on the relative robustness of subjective vs. objective measures).

- The pushback on Ruggeri’s ‘positive deviance’ findings is interesting, but it does not address his work that finds biases or less ‘rational’ temporal discounting choice is not the purview only of those experiencing financial hardship (see Ruggeri et al., 2022).

6. PLOS authors have the option to publish the peer review history of their article (what does this mean?). If published, this will include your full peer review and any attached files.

Reviewer #1: No

Reviewer #2: **Yes: **Julia Buzan & Jennifer Sheehy-Skeffington

---

## [Author Response · Author response to Decision Letter 0]

31 May 2024

Dear Dr. Akaishi,

Thank you very much for the thoughtful critiques that you solicited, and that the reviewers provided for this project. We are very encouraged that the reviewers saw merit in our work. We have revised the manuscript to address all of the points raised in the reviews. To facilitate your evaluation of this revised manuscript, we have reproduced excerpts from each critique in italics, followed by our responses

Reviewer #1

● The manuscript by Wang and Hanson provides an interesting test of how childhood poverty can influence adult decision-making, particularly temporal discounting

We thank the reviewer for these positive remarks!

● My main concern about the paper can also be considered a strength [...] Large datasets are great, and the use of a multinational dataset increases the generalizability of the data. However, what was the effect size? How important is this relationship given the very small beta coefficients? This does not detract from the author’s ideas, but I believe it is critical to understand the importance of these previous experiences to current decision making.

This is an important point raised by this reviewer. In our revised manuscript, we have tried to provide a much more tempered discussion about the strength of these relations throughout the document. Furthermore, in response to this comment, we have included specific effect size estimates (e.g., Pg. 13), as measured by Cohen’s d, for different associations noted in our results section. If there is a preference for a different statistical parameter estimate (Hedges’ g; partial eta), we are happy to include those in future versions of the manuscript.

In addition, given the fact that there are many intervening factors from childhood to adulthood, we believe it is still notable to find relations between childhood SEP and adult temporal discounting. Within our sample, heterogeneous subgroups likely exist, and it would be very interesting to explore if childhood SEP interacted with these intervening factors to predict later adult behavior. We think this is a great direction to pursue in future projects. However, noting these long-term influences on behavior is an important, initial contribution to the literature. 

● In summary, I really liked this paper, but the authors need to indicate the importance of these results. In the beginning of the results, the authors describe the relationships as modest. That is vague. The important point, if possible, is to Even if small, this may be very important and definitely worth reporting. I look forward to reviewing a revised version of this stimulating paper.

We thank the reviewer for raising this concern. We have added further discussion of this issue in our results section (e.g., Pg. 13). 

Reviewer #2

● The paper is an interesting discussion of an important topic, but as it stands I am not sure I would recommend the current paper for publication for three reasons: First, it does not lay out a clear enough theoretical framework through which to guide the research’s approach, predictions, and findings. Although theoretical contribution is not a key criterion for publication in PLOSOne, it does matter to the extent that the questions and the way of approaching them can be scientifically justified and easily interpreted.

● The framing does not enough outline why individual-level decision-making should necessarily be implicated as an important topic of study in the association between experiences of adversity and temporal discounting (as opposed to say structural disadvantage, e.g., lack of access to healthcare), and it does not yet clearly enough argue for why and how experiences of adversity in childhood should affect decision-making, above and beyond current experience.

We thank the reviewer for these prompts for improvements. We believe this is clear and directive feedback and highlights a past limitation of the manuscript. In response to this comment, we have expanded the framing and introduction to better conceptualize individual decision-making as an important, though not sole, mechanism in the relations between socioeconomic disadvantage and later outcomes. Most notably, we highlight how socioeconomic disadvantage: 1) is associated with a myriad of other challenges and life stressors (lines 47-58); and 2) can influence behavior and brain development, influencing reward processing, executive functions, and their connected neural circuits (lines 81-95). We hope that this more richly motivated the work and improves the reviewer’s opinion of the manuscript. Of note, we did not provide an exhaustive review of these research areas as we did not actually have measurements of these constructs (e.g., stress exposure; fMRI responsivity/neurobiology); however, we are happy to further expand this framing if the reviewer thinks appropriate.

● Despite framing the gap being addressed as a lack of understanding of the link between SEP and specific decision-making processes, the temporal discounting items used are undertheorized and underexplained – what is different in the gain, loss, and magnitude frames, and why are all three included? Why should we care about current income more so than other economic aspects of financial adversity (as Ruggeri et al., 2022’s paper on temporal discounting indicates we should)? What are the specific ‘choice anomalies’ tested for in the supplemental questions, and what can they tell us above and beyond the primary measures? How were the income titration percentages calculated, and why do they seem to use both participant income and national household average income? Why can we be confident in grouping together all of the measures into a single average (and is an average appropriate – i.e., would a factor analysis allow for a more appropriate weighting of the different components?) More theoretical contextualization is needed to explain why each temporal discounting measure should be included, what we should expect from each, and why they should be combined.

We thank the reviewer for raising these issues. There are elements that we strongly agree with. There is also feedback that we believe provides an opportunity to clarify or strengthen motivation. Related to the primary measure of discounting that we used, we believe it is important to note that this is one of the first projects, to our knowledge, to examine delay discounting in adulthood and correlate it with childhood socioeconomic position. Given this fact, we believed that our results would most cleanly connect to the large body of delay/temporal discounting work that has looked at these measures in the aggregate (similar to our measures) and not with a more granular focus on choice anomalies. In addition, we have added clarifying language in the Methods section regarding the grouping of the different constructs and how the score was calculated (Pgs. 9-10). We hope these additional details will help to explain the robustness of the temporal discounting measure and address the reviewer’s concerns about the appropriateness of this measurement technique. 

● More attention could be paid in the initial framing of subjective vs. objective measures of childhood SEP. The measure of adult SEP is particularly problematic, as judgment of how things have changed cannot be used as a proxy for judgment of how one is doing overall (as it can make a poor person who did well in the pandemic look like they are higher in SES than a rich person who did poorly in the pandemic). The wording of this measure thus needs to be changed throughout, to something like ‘recent economic change’; alternatively, so that they can speak to something like adult SEP, the authors could use adult income in their analyses instead.

We thank the reviewer for calling our attention to these areas for improvement and the lack of clarity on our description of childhood SEP measures. In connection to this comment as well as others raised by the reviewer about “objective” measures of childhood SEP, we have decided to move this set of analyses to the supplement as comparing objective and subjective measurements was not a main focus of the study. Regarding the wording of the adult SEP variable, we agree with the reviewer’s assessment and elected to change the variable name to recent economic change as suggested. 

● It seems that one of the key arguments is to push back on Ruggeri’s ‘positive deviance’ findings, but this is not mentioned in the setup. I would love a clearer articulation of the study’s theoretical framework, subsequent predictions, and if/how those predictions might prove important given the context of alternate findings in the literature.

We thank the reviewer for noting this opportunity to strengthen the theoretical framework of the study and have done a complete restructuring of the introduction section to more clearly present the theory motivating our hypotheses. Some language remains the same, but we have included additional paragraphs (in lines 81-111) to further articulate the study’s theoretical framework. 

● Although preregistration of analysis of secondary data is only now becoming a norm, its absence here makes it harder to both 1) interpret how the analyses link to the theoretical framing, and 2) discount the role of researcher degrees of freedom. (E.g., a global temporal discounting score that combines the sub-variables would be more convincing if this analysis approach had been pre-registered. Without the pre-registration and especially without documentation of the related code readers are left wondering if the results hold only when the sub-items are combined).

We appreciate this feedback and thought. However, given that analyses have already been completed, we are unsure how best to respond to this point. Our lab is doing more pre-registration for different studies, but the reviewer is correct that the project was not pre-registered. We have more clearly noted this on Pg. 6.

● Relatedly, the lack of supplemental information means that the analyses reported cannot be checked, and greater detail on the measures and study sample used cannot be adequately considered.

We apologize for this omission, we did not realize this was a necessary component for submissions to PLOS-one. In the revised submission, we have included additional details about measures and items, as well as data analysis. 

● Similarly, a greater discussion could be included as to why each method was selected, the reasoning behind how each method was conducted (e.g., more detail on the reasoning behind country-level random effects), and more reporting of missing values (why are missing values imputed?) and related robustness checks.

We thank the reviewer for bringing this to our attention and agree that further reasoning for the use of statistical models is warranted. We have added clarifying language on Pg. 12 to explain why we chose linear mixed-effects models as well as the reasoning for including country as a random effect in the models. Furthermore, we have included additional information regarding our exploratory analyses included in the supplement. Regarding the use of imputation, we have moved this set of analyses to the supplement as referenced in an earlier comment.

● As I understand it, the dataset being used was collected by other researchers, however this should be more clearly outlined (and the benefits and limitations discussed). It is also not clear from the original authors’ OSF file whether analysis of this data for use in future publications (as opposed to just for replication of the analyses in their paper) is permitted.

We apologize for the lack of clarity here. We believe we noted this in our original submission, but the reviewer is likely correct that it is a detail that may be missed by readers. We have now revised the manuscript to more specifically emphasize this fact. In particular, a paragraph was added to the discussion (lines 377-387). 

● The use of existing datasets can be sufficient to warrant publication, however the current analyses underleverages the potential of the existing dataset. (E.g., getting beyond WEIRD samples is noted as a benefit in the Discussion section, but no analysis is made to consider if and how results vary according to country context such as country GDP and inequality).

We appreciate this feedback and thought. In our supplemental models, we have now included a number of additional analyses including breaking effects down by individual country, comparing the results of our models in WEIRD versus non-WEIRD countries, and a strengthened explanation of the country-level analysis of global extreme poverty rate from the original manuscript. We believe these elements begin to unpack how country/context may influence our results. Of important note (in our supplement), we examined relations between discounting and childhood SEP in each country separately, running 61 statistical models with childhood SEP as the independent variable and discounting as the dependent variable. In these analyses, we find that participants in ~95% of the countries surveyed showed this association (see Supplement Pg. 2 and Figure S3). 

● If the paper is specifically pointing to limitations in existing research (Ruggeri’s ‘positive deviance’ findings), then it could more tightly point to limitations in that existing research and structure the analyses and discussion appropriately – as it stands I worry it is both too specific and too broad to meaningfully contribute above and beyond the original analysis.

We appreciate the reviewer for raising this comment. In response to this critique, we had added discussion of Ruggeri’s ‘positive deviance’ findings (on Pg. 18). Specifically we noted that “we analyzed the full range of childhood socioeconomic positions [..] Decision-making processes may continue to be negatively affected by childhood experiences well into adulthood, affecting responses to negative changes in one’s economic situation. This may be true even among those whose financial situation has improved since their early years”. We believe this addition helps contextualize our findings in relation to past work by Ruggeri et al., but also a much larger body of research that looks at childhood SEP as a driving factor of disparities. This latter work does not typically focus on “deviances” between early and later social position, instead arguing that early social position directly and uniquely shapes later behavior, irrespective of later positioning and change.

● While Figure 3 is compelling, greater discussion is needed to explain differences in the relationship between childhood SEP when measured continuously or discretely (Figure 4).

● As outlined in the discussion, the paper does not dig deeply into underpinning mechanism. As such it is hard to rule out alternative explanations, particularly given the cross-sectional nature of how the data are reported.

We thank the reviewer for this comment. As noted above, we have expanded our discussion into how socioeconomic disadvantage: 1) is associated with a myriad of other challenges and life stressors; and 2) can influence behavior and brain development, influencing reward processing, executive functions, and their connected neural circuits. We believe this begins to move closer to mechanism; however, we want to be careful as to not go too far into this space (as we did not actually have measurements of these constructs). We have revised our introduction to help further motivate and contextualize the work (Pgs. 4-5). Regarding relations between childhood SEP when measured continuously or discretely, we have added a brief discussion in our manuscript’s supplement (Supplement Pg. 9). Specifically, we note that: “we see that the lowest childhood SEP group shows greater temporal discounting than the multiple other groups (below average, above average). However, depending on the model covariates, there were variations in differences between the poor and other groups. This suggests that a continuous measure of social position may obscure non-linear relationships that become evident when analyzing this variable categorically. The variability in findings for the Around average v. Poor groups are surprising and do not want to overinterpret this null effect. For the hig

---

## [Decision Letter · Decision Letter 1]

22 Jul 2024

Childhood socioeconomic position related to adult decision-making: Evidence from a large cross-cultural investigation

PONE-D-23-32303R1

Dear Dr. Hanson,

We’re pleased to inform you that your manuscript has been judged scientifically suitable for publication and will be formally accepted for publication once it meets all outstanding technical requirements.

Kind regards,

Rei Akaishi

Academic Editor

PLOS ONE

Additional Editor Comments (optional):

This paper is basically accepted. Please respond to the comments of Reviewer 1: "I would prefer the limitation of small effect size to be upfront in the discussion, rather than the last point."

Reviewers' comments:

Reviewer's Responses to Questions

**Comments to the Author**

1. If the authors have adequately addressed your comments raised in a previous round of review and you feel that this manuscript is now acceptable for publication, you may indicate that here to bypass the “Comments to the Author” section, enter your conflict of interest statement in the “Confidential to Editor” section, and submit your "Accept" recommendation.

Reviewer #1: (No Response)

Reviewer #2: (No Response)

2. Is the manuscript technically sound, and do the data support the conclusions?

Reviewer #1: Partly

Reviewer #2: Yes

3. Has the statistical analysis been performed appropriately and rigorously? 

Reviewer #1: Yes

Reviewer #2: Yes

4. Have the authors made all data underlying the findings in their manuscript fully available?

Reviewer #1: Yes

Reviewer #2: Yes

5. Is the manuscript presented in an intelligible fashion and written in standard English?

Reviewer #1: Yes

Reviewer #2: Yes

6. Review Comments to the Author

Reviewer #1: My major concern is still a concern. The magnitude of the relationship is very small, with very small effect sizes. The significance is due in part to the large sample size. This is not mentioned at all in the limitations section, but it should be upfront. The only effect that is reported to be large is the effect reported on p 13 of the revised manuscript where the model accounted for 22% of the variance. The difference between the models based on the type of analysis also should be noted in the discussion. I would prefer the limitation of small effect size to be upfront in the discussion, rather than the last point.

I recognize that adult reporting on childhood experiences is often biased, and not just because of memory issues, which are real, but rather interpretation in light of the past based on current experience. the authors can not do more than note this problem.

Reviewer #2: My overall sense is that the authors took our comments seriously and that they significantly improved the manuscript. As such, I think the piece now meets the bar for PLOS ONE, and is ready for publication pending consideration of a few minor recommendations.

Theoretical framework and motivation

• This is now much clearer.

• The reader still needs to make a bit of a leap between the stress literature and the specific domain of temporal discounting.

o I would recommend a little more linking stress to temporal discounting (paragraph starting line 96), i.e., Why should we assume that a stressful event will result in higher levels of temporal discounting? More clearly citing previous findings in the literature (or even a sentence that more clearly spells out the guiding logic) would be helpful.

Similarly, in the conclusion the attribution to stress (317) may be premature, i.e., stress is a potential (but untested) explanatory mechanism.

o The authors make some (I believe unintended) normative judgements on temporal discounting being necessarily evidence of something negative (110). Seen through another lens, temporal discounting could, in fact be adaptive (e.g., if I grew up in a harsher environment, and if Covid just affected me negatively then I may have reasonable experiential grounds upon which to determine that waiting for delayed rewards may not be the most ‘optimal’ course of action, given my environment). The authors gesture at this in the discussion when they point to the potential role of volatility (365), but the authors could consider if and how temporal discounting (in an environment characterized by uncertainty, unreliability, and discrimination) is necessarily ‘suboptimal’ a conclusion that is challenged by work that considers decision-making in adversity from an adaptive perspective (see Sheehy-Skeffington, 2020, for a review). Research taking the latter perspective also features two empirical studies testing a similar form of stress sensitization as is posited here, examining how childhood adversity interacts with current economic strain to shape present-biased decisions (Griskevicius et al., 2014) and how childhood unpredictability (a factor mentioned in the discussion of the current manuscript) interacts with current unpredictability to shape inhibitory control (Mittal et al., 2014). As both papers involve both outcome variables closely connected to temporal discounting, and look at interactions between childhood and recent socioeconomic experiences, it’s important they are cited in the manuscript with its current theoretical framing.

Methodological transparency

• The piece now goes much farther in documenting the methodologies employed (and the reasoning behind why they were used).

• I still believe the discussion (378) could benefit from greater reflection on limitations in the measures used (specifically the possibility for method variance in pessimism to be driving results). Here it could be that respondents who are more pessimistic are more likely to rate both their childhood and their Covid experience negatively, that pessimistic expectations about the future drive higher levels of temporal discounting, rather than ‘objective’ experiences of adversity.

Contribution

• The contribution remains modest (given limitations outlined in the previous review, e.g., the focus on summary measures only rather than a more detailed examination of sub-measures, limited insight into mechanism, limitations of subjective vs objective measures, etc), but I think it now meets the bar for publication at PLOS ONE

Works cited:

Griskevicius, V., Ackerman, J. M., Cantú, S. M., Delton, A. W., Robertson, T. E., Simpson, J. A., ... & Tybur, J. M. (2013). When the economy falters, do people spend or save? Responses to resource scarcity depend on childhood environments. Psychological Science, 24(2), 197-205.

Mittal, C., Griskevicius, V., Simpson, J. A., Sung, S., & Young, E. S. (2015). Cognitive adaptations to stressful environments: When childhood adversity enhances adult executive function. Journal of Personality and Social Psychology, 109(4), 604.

Sheehy-Skeffington, J. (2020). The effects of low socioeconomic status on decision-making processes. Current Opinion in Psychology, 33, 183-188.

7. PLOS authors have the option to publish the peer review history of their article (what does this mean?). If published, this will include your full peer review and any attached files.

Reviewer #1: No

Reviewer #2: **Yes: **Julia Buzan & Jennifer Sheehy-Skeffington

---

## [Editor Report · Acceptance letter]

11 Sep 2024

PONE-D-23-32303R1 

PLOS ONE

Dear Dr. Hanson, 

I'm pleased to inform you that your manuscript has been deemed suitable for publication in PLOS ONE. Congratulations! Your manuscript is now being handed over to our production team.

Kind regards, 

on behalf of

Dr. Rei Akaishi 

Academic Editor

PLOS ONE